# Altitude Modifies the Effect of Parity on Birth Weight/Length Ratio: A Study Comprising 2,057,702 Newborns between 1984 and 2020 in Austria

**DOI:** 10.3390/life13081718

**Published:** 2023-08-10

**Authors:** Eva Karner, Dana A. Muin, Katrin Klebermass-Schrehof, Thomas Waldhoer, Lin Yang

**Affiliations:** 1Division of Feto-Maternal Medicine, Department of Obstetrics and Gynecology, Medical University of Vienna, 1090 Vienna, Austria; eva.karner@meduniwien.ac.at (E.K.); dana.muin@meduniwien.ac.at (D.A.M.); 2Division of Neonatology, Pediatric Intensive Care and Neuropediatrics, Department of Pediatrics, Medical University of Vienna, 1090 Vienna, Austria; katrin.klebermass-schrehof@meduniwien.ac.at; 3Department of Epidemiology, Center for Public Health, Medical University of Vienna, 1090 Vienna, Austria; 4Department of Cancer Epidemiology and Prevention Research, Alberta Health Services, Calgary, AB T2S 3C3, Canada; linyang33@gmail.com; 5Departments of Oncology and Community Health Sciences, University of Calgary, Calgary, AB T2S 3C3, Canada

**Keywords:** altitude, birth weight, parity, birth weight/length ratio, length, population-based study

## Abstract

(1) Background: Lower birth weight among newborns in higher altitudes has been well documented in previous literature. Several possible causes for this phenomenon have been investigated, including biophysiological adaptation, epigenetic or genetic mechanisms or lifestyle changes. This is the first study to show the effect modification of altitude and parity on the birth weight length ratio (BWLR) in women resident in moderate altitudes compared to a low sea level.; (2) Methods: This population-based study obtained data on altitude (0–300, 300–500, 500–700,700–900, >900 m), parity (1, 2, …, 7, 8/9), birth weight and length on all births in Austria between 1984 and 2020 from birth certificates provided by Statistics Austria. The BWLR was calculated, and the effect of moderate altitude and parity was estimated using multivariable linear mixed models adjusting for predefined variables. Sub-group regression analyses were conducted by altitude group. (3) Results: Data on 2,057,702 newborns from 1,280,272 mothers were analyzed. The effect of parity on BWLR, as indicated by the difference of BWLR between the first- and second-born infants, ranged between 1.87 to 2.09 g per centimeter across all altitude groups. Our analyses found that the effect of parity on BWLR diminished from parity three onwards at altitude 0–300, whilst the effect of parity on BWLR continued to increase at higher than 300 m and was most notable in the highest altitude group >900 m. (4) Conclusions: Findings from our study indicated that the negative effect of increasing altitude on BWLR was deprived for newborns of higher parity. It shows that the residential altitude can modify the effect of parity on BWLR.

## 1. Introduction

Low birth weight and fetal growth retardation are associated with higher risks of morbidity and mortality in newborns [1]. In addition, fetal growth retardation is associated with negative outcomes later in childhood through adulthood, including higher risks of bronchial asthma, various metabolic diseases and cardiovascular dysfunction [2,3,4,5], and impaired cognitive and emotional development, as well as physical skills [3,6]. Due to the increased morbidity, there is an overall reduced health-related quality of life for those children [7]. Factors compromising low birth weight include fetal sex, young or advanced maternal age, sociological status, smoking, malnutrition and primiparity [8,9,10]. High altitude poses a negative effect on fetal birth weight with a decreased birth weight of 100–150 g per 1000 m altitude elevation [11].

Altitude is not only described to have an influence on birth weight but also on fetal length. As asymmetric growth describes the different increases in head circumference and abdominal circumference, the birth weight/length ratio serves as a valid predictor for neonatal outcomes [12,13]. Evidence from our previous longitudinal study analyzing the birth weight of siblings indicated that the effect of moderate altitude on fetal growth appeared to be causal and that birth weight increased with parity. Nevertheless, the effect of parity and altitude on birth weight length ratio (BWLR) has not been investigated [14]. Several hypotheses have been proposed to explain the cause of the effect of altitude on birth weight. Physiologic adaptation mechanisms in maternal biomolecular pathways, including oxygen deprivation and glucose supply, have been discussed [15,16,17]. Other studies suggest epigenetic mechanisms as a result of adaptation processes [18,19]. However, a positive effect on birth weight is described for multiparity [20], such that a significant increase in birth weight with parity has been observed. A particular increase in birth weight was described between first- and second-born siblings [8,21,22]. The aim of this work was to investigate the influence of moderate altitude on the effect of parity on the birth weight/length ratio.

## 2. Materials and Methods

Data comprise all birth certificates in Austria between 1984 and 2020 (Statistics Austria) and has been described in detail previously [14]. All data were retrieved anonymously.

Altitude, in meters (m), was determined by the centroid of the municipality according to mothers’ living addresses. Birth weight was measured in grams. Birth weight was recorded in intervals of 100 g, 10 g, and 1 g from 1984 to 1998, from 1999 to 2010 and from 2011 onwards, respectively. Birth length was measured in centimeters (cm).

The following information was used to define inclusion criteria: Austrian citizenship of the mother, maternal age 15–50 years, live birth, gestational age 30 to 44 weeks, birth weight >1000 g, single birth and parity between 1 and 9.

The final analysis included mothers whose living address (grouped into five altitude groups: 0–300, 300–500, 500–700,700–900, >900 m) remained the same over the observation period. This approach avoided cross-over effects when mothers giving birth to more than one newborn moved between altitude levels.

### Statistical Analysis

The birth weight/length ratio (BWLR in g/cm) was calculated. The effect of altitude on BWLR was estimated using multivariable linear mixed models in SAS (procedure HPMIXED) adjusting for the following variables: education of mother (categories: completed compulsory school; apprenticeship; school for intermediate vocational education; academic secondary school, college for higher vocational education, university), sex, altitude of residence (kilometers), gestational age (weeks), gestational age squared, maternal age (years), year of birth of the newborn and parity (1, 2, …, 7, 8/9). Parity 8 and 9 were grouped into one single category in order to maintain a sufficient number of observations within each parity group. Newborns of parity 10 and more were excluded because of too few cases.

An indicator for the mother was entered as a random effect with an autoregressive correlation term of the first order.

In the first step, the interaction between altitude and parity was tested and turned out to be highly significant (*p* < 0.001). Therefore, we re-ran the regression model within each sub-group of altitude categories (0–300, 300–500, 500–700,700–900, >900 m) to estimate the effect of parity on BWLR in dependence of altitude. Altitude in meters was kept in the sub-group-specific regression analysis as a continuous variable to avoid residual confounding.

The combined effect of altitude and parity on BWLR was depicted by a figure showing the estimated difference and 95% confidence intervals of BWLR compared to the first newborn’s BWLR.

The absolute effect of altitude and parity was depicted by showing predicted BWLR and corresponding 95% confidence intervals for a newborn with the following characteristics: male, born in the 39th week of gestation in the year 2000, mother aged 30 years with the educational level of “school for intermediate vocational education”.

Descriptive statistics for BWLR in dependence of altitude and parity are estimated using means and 95% confidence interval, and 1st, 2nd and 3rd quartiles (Q1, Q2, Q3). All analyses were performed using SAS 9.4 (SAS Institute Inc., Cary, NC, USA).

The study was approved by the Ethics Committee of the Medical University of Vienna and complied with the principles outlined in the Declaration of Helsinki of 1975, as revised in 2013. Participants’ written consent was not required per the Austrian Federal Act concerning the Protection of Personal Data (DSG 2000). All individual data were de-identified before analyses.

## 3. Results

A total of 2,231,866 observations initially entered the analysis, from which 174,164 observations (8.4%) were removed because mothers moved between altitude levels. Therewith, 2,057,702 newborns from 1,280,272 women remained in the analysis. The mean altitude of mothers’ residential addresses was 430 m (Q1 = 240, Q2 = 400, Q3 = 550) and a minimum of 120 m and a maximum of 1666 m for the whole study population. Overall, 37.2% of the first newborns were delivered at the lowest altitude group (0–300 m), 3.8% at the highest altitude (>900 m) (see Appendix A), and mothers residing at the lower altitudes had higher education levels than those residing at the higher altitudes (see Appendix A).

The mean BWLR was 66.1 g/cm (Q1 = 61.5, Q2 = 66.3, Q3 = 71.0). The mean BWLR was 66.45 g/cm (66.43, 66.47) at the lowest altitude (0–300 m) and decreased to 64.59 g/cm (64.54, 64.64) at the highest altitude group (>900 m).

The mean BWLR was 64.90 g/cm (64.88, 64.91) for first newborns and increased to 67.35 g/cm (66.88, 67.81) for eighth/ninth newborns (see Appendix A).

The results of the regression model can be seen in Appendix A.

Figure 1 presents the results obtained from the regression model, depicting the combined effect of altitude and parity on BWLR. Specifically, it illustrates the estimated difference between the birth weight-to-length ratio (BWLR) of later-born newborns and first-born newborns’ BWLR in each attitude subgroup.

The BWLR of second-born infants is higher than that of first-born siblings by 1.87 to 2.09 g per centimeter across all altitude sub-groups (see Appendix A). This difference in BWLR is the smallest among newborns whose mothers reside at altitudes > 900 m and the largest among newborns at altitudes < 500m.

Differences in BWLR compared to first-born infants gradually increases from parity 2 to 8–9 in all altitude groups. For newborns at the lowest altitudes (0–300 m), the increase in BWLR from parity 3 onwards is minimal, indicating the effect of parity stays constant. This is in contrast to all other altitude groups, where notable increases in BWLR between the third and fifth birth were observed, i.e., higher birth order predisposes to a greater difference in BWLR in-between siblings.

In particular, the increase in BWLR was observed from parity 3 to 5 and then remained stable in altitude groups 300–500 m and 700–900 m, whereas the increase in BWLR continued from parity 3 up to parity 8–9 in altitude groups 500–700 m and >900 m.

It is important to note that the number of newborns per parity decreases in the population, leading to increased uncertainty in the observed effects of parity as indicated by wider 95% confidence intervals. This trend becomes pronounced from parity 6 onwards, limiting the statistical power in comparisons between altitude groups.

Figure 2 displays the combined effect of parity and altitude on BWLR using a hypothetical newborn with the following characteristics: male gender, born in the 39th week of gestation in the year 2000, with a mother aged 30 years and an educational level of “school for intermediate vocational education”.

In the altitude group of 0–300 m, the plot reveals a minimal increase in BWLR for siblings with parity of three or more. However, in all other altitude groups, the BWLR shows a progressive increase as parity increases.

## 4. Discussion

Our longitudinal study showed increasing BWLR with increasing parity, particularly among women residing at higher altitudes. The detrimental effect of increasing altitude on BWLR was weakened for newborns of higher parity, and the increase in BWLR for higher parity was considerable for mothers residing at moderate altitudes > 900 m.

Asymmetric growth restriction at higher altitudes has been well documented in the literature [12,16]. Newborns experience reduced birth weight but greater abdominal and head circumferences [23]. Therefore, the BWLR can be used as a better comparable parameter for neonatal outcome than birth weight alone. Our study observed that the BWLR increases with parity, and this increase was modified by altitude. However, BWLR increases continuously in newborns at higher altitudes yet stagnates at low altitudes in subsequent pregnancies. Different causes for the effect of altitude on fetal growth have been discussed [16]. Chronic hypoxia in pregnant women residing at high altitudes may lead to alterations in uteroplacental vascularisation [13]. Living in hypoxic conditions stimulates the expression of hypoxia-inducible-factor-1 and -2 (HIF-1, HIF-2), which seem to interact with other angiogenetic factors, such as vascular endothelial growth factor (VEGF) and Placental growth factor (PlGF) [24,25]. Both factors play major roles in the placentation process and the maintenance of the placental function, which consists primarily of the oxygenation and nutrition of the fetus [19,26]. Other authors also suggest genetic or epigenetic influences operating the effect of altitude on fetal growth [18,27].

The increase in birth weight with parity has been described for years, where the existing literature only addresses birth weight but not BWLR. Primiparous women are more likely to deliver newborns with lighter birth weight and a higher risk of being small for gestational age (SGA) than multiparous [28]. In general, multiparity is associated with more favorable obstetric outcomes. Maternal changes between pregnancies were investigated as a possible cause of the effect of parity. Changes, such as weight gain, changing medical conditions, and higher maternal age in subsequent pregnancies were discussed but were not associated with significant influence on the increased birth weight with parity. Stress and maternal physical activity have not been investigated in the referred studies [21,29]. Another possible explanation for this effect is the physiological change in uterus anatomy, size, and arterial innervation during and after the first birth [30]. Autopsy studies of the 19th century showed a weight gain of the postpartum uterus and also an augmentation of the uterine size with parity [31]. A sonographic study of the uterus complies with the statement and describes a significant increase in uterine cavity size in multiparas [32]. According to Hermanussen, cavity capacity is linked to birth weight [30]. Not only the uterine size but also the placenta size increases with parity, as demonstrated by a large Scandinavian study by Flatley et al. [33] Goldman-Wohl et al. discussed improved maternal revascularisation and biological memory as the basis of the remodeling of uterine spiral arteries [34]. Histologic studies also showed structural changes in the architecture of the uterine artery wall. Based on these findings, Khong et al. presumed that these structural modifications prepared the uterine vessels for subsequent pregnancies. Accordingly, the implantation of the trophoblast and the process of remodeling the uterine spiral arteries are more efficient in subsequent pregnancies, especially after the first pregnancy [35]. Another explanation for the effect of parity on birth weight is the presence of natural killer cells in the decidua. Chemokines, growth factors and angiogenic factors produced by natural killer cells are important for placentation and affect the endothelial cells of the spiral arteries. Gamliel et al. described memory in the immune system due to the expression of antigen-receptors by the natural killer cells, which leads to cellular support of subsequent pregnancies [34,36].

Our study now demonstrated a positive effect of increasing parity on BWLR in women living at higher compared to lower altitudes.

Potential mechanisms leading to the influence of altitude on the parity and BWLR association can only be speculated. The chronic hypoxic situation in pregnant women seems to have an effect on the biomolecular pathways of the placentation. Knowledge of biomolecular pathways in the physiological adaption process of pregnant women in hypoxic environmental conditions can lead us to important information about the pathophysiological changes in placental insufficiency and the development of fetal growth retardation. Insights into these mechanisms subsequently lead to the establishment of new possibilities in the prevention and management of placental insufficiency and following conditions [15]. In view of previously acknowledged physiological mechanisms of the placentation in parous uteri, additional hypotheses can be postulated. Most probably, the remodeling of spiral arteries is a key element in the determination of fetal growth. On the other hand, placentation is completed within 24 gestational weeks, whereas studies show that the negative effect of altitude on birth weight exhibited the most progression in the third trimester of pregnancy [37,38,39]. Further studies are needed to confirm and strengthen our findings. It is unclear whether lifestyle habits as per altitude contribute to the differences in BWLR. After all, we can assure homogenous medical and fetal care throughout pregnancy across the country. However, possible influences could be stress, nutrition or physical activity in pregnant women as per altitude. As our study does not have information on these characteristics, we cannot investigate associations with these potential influencers. Our previous longitudinal population-based study investigated the effect of altitude on birth weight within siblings of the same mother who migrated across altitude groups between two births. The findings showed decreased birth weight in the second birth when the mother moved from a higher altitude to lower compared to those who moved up. Gene-environment interactions have been discussed, but it is also possible that lifestyle habits as per altitude may also contribute to the difference in fetal growth [14]. However, it is worth mentioning that the previous study investigated the effect of moderate altitude on birth weight but not on BWLR. Research on the effect of altitude on morbidity and mortality shows favorable effects of high altitude on cardiovascular diseases and less favorable effects on chronic pulmonary diseases in the general population [40]. Overall, more health benefits are described for people living at moderate altitudes as a result of environmental conditions, including hypoxia and climate differences as well as lifestyle behavior [41,42]. How lifestyle behavior at higher altitudes affects pregnant women should be further investigated.

Our study contributes an important part in the investigations of the effect of moderate altitude on BWLR, owing to the longitudinal study design covering the first and later borns of the same mother. In addition to the study design, our study examined BWLR, which carries higher predictive power regarding perinatal outcome than birth weight alone. The major limitation of our study is the lack of information regarding obstetric characteristics. No data is available on the distribution of maternal or fetal morbidities of pregnant women. In particular, the occurrence of pregnancy-associated hypertensive disorders or gestational diabetes, as well as other metabolic or cardiovascular morbidities, would be potential factors to evaluate possible physiological or pathophysiological pathways. Because the occurrence of lower birth weight is more frequent in fetuses with chromosomal anomalies, the exclusion of pregnancies with fetal chromosomal anomalies would be useful in the evaluation of external effects on BWLR. Finally, information on the personal lifestyles of the pregnant population was not available and thus limited the possibility of investigating whether and how environmental and lifestyle factors may influence the effect of altitude on fetal BWLR.

In conclusion, our study adds knowledge by investigating the potential effect modification of altitude on the parity and BWLR association in Austria and shows a stronger effect of parity on BWLR at higher altitudes.

## Figures and Tables

**Figure 1 life-13-01718-f001:**
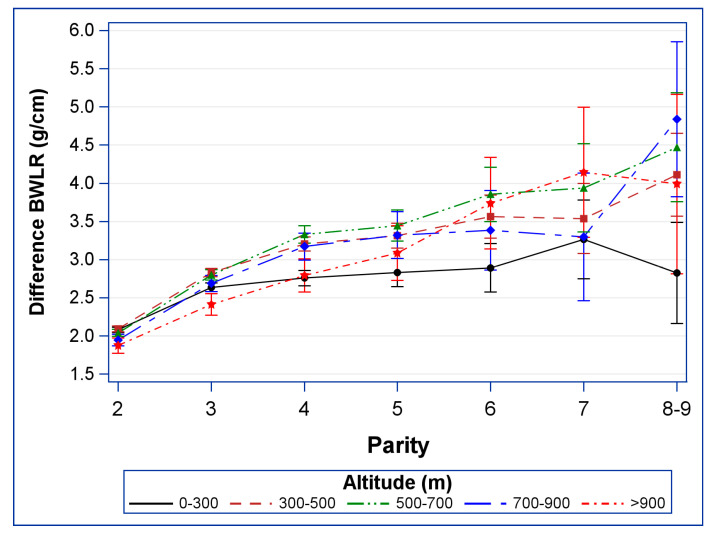
The relationship between altitude, parity, and the difference in BWLR between first-born and later-born infants.

**Figure 2 life-13-01718-f002:**
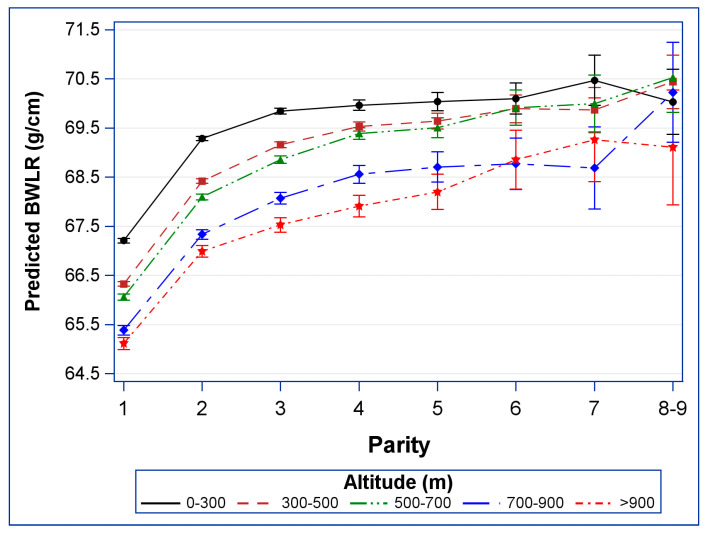
The estimated BWLR for a hypothetical newborn as described in the last paragraph of the methods section.

## Data Availability

Restrictions apply to the availability of these data. Data was obtained from Statistics Austria and are available from Statistics Austria with the permission of Statistics Austria.

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
