# Peer review of "Altitude Modifies the Effect of Parity on Birth Weight/Length Ratio: A Study Comprising 2,057,702 Newborns between 1984 and 2020 in Austria"

_life, 2023, doi:10.3390/life13081718_

Round 1

Reviewer 1 Report

The paper is well written and appears to be competently analyzed but I have four major, and two minor concerns.

Major:

1. The authors are overstating their case. The title, abstract, introduction and elsewhere refer to "high altitude" (or "altitude" which most readers will interpret as "high" altitude) whereas theirs’s is a study of moderate altitude. There is very little decline in SaO2 at the altitudes considered! While pO2 declines more or less linearly, SaO2 does not due to O2-Hb binding kinetics.

2. They seem to be milking their dataset to generate multiple, related papers (i.e., taking the “mpu or minimal publishable unit” path). Is that truly justified or necessary?

3. This is not the first study to consider the effect of parity; see for example Jensen 1997 (Am J Pub Health)

4. Does Austria not link infant death and birth weight records? If so, have the authors considered the mortality impact of the truly modest reductions in birth weight being reported? The 1st line of the introduction (and elsewhere) stress the importance of birthweight or the birth weight length ratio for survival. But is there any effect at these moderate altitudes?

Minor:

1.     Please add a reference for the 100-150 birth weight decline per 1000 m altitude gain, which is Jensen and Moore, 1997

2.     Line 250: I think you mean “maternal” not “material”

none

Reviewer 2 Report

Low birthweight and fetal growth retardation are associated with higher risks of morbidity and mortality in newborns. Altitude is not only described to have an influence on birthweight, but also on the fetal length. The aim of this study was to investigate the influence of altitude on the effect of parity on birth-weight/length ratio (BWLR). The authors showed that the effect of parity on BWLR, as indicated by the difference of BWLR between the first- and second-born infants, ranged between 1.87 to 2.09 grams per centimeter across all altitude groups. Furthermore, the effect of parity on BWLR diminished from parity three onwards at altitude 0-300. The authors conclude that this study adds knowledge by investigating the potential effect modification of altitude on the parity and BWLR association in Austria, and shows a stronger effect of parity on BWLR at higher altitude. The manuscript is well-written and the methods sound. I did not have any major concerns, only minor issues listed below:

Page 5 lines 160, “Figure 1. The estimated BWLR for a hypothetical newborn, as described in the last paragraph of the methods section.” This legend is Figure 2. I could not find the description in the last paragraph of the methods section. Please describe in detail here.
